rsob.royalsocietypublishing.org

**Subject Area:**
developmental biology

*Drosophila*, embryogenesis, maternal-to-zygotic transition, zygotic genome activation, Zelda, cellular reprogramming

**Author for correspondence:**
Melissa M. Harrison
e-mail: mharrison3@wisc.edu

# Regulatory principles governing the maternal-to-zygotic transition: insights from *Drosophila melanogaster*

Danielle C. Hamm and Melissa M. Harrison

Department of Biomolecular Chemistry, University of Wisconsin School of Medicine and Public Health, Madison, WI 53706, USA

 MMH, 0000-0002-8228-6836

The onset of metazoan development requires that two terminally differentiated germ cells, a sperm and an oocyte, become reprogrammed to the totipotent embryo, which can subsequently give rise to all the cell types of the adult organism. In nearly all animals, maternal gene products regulate the initial events of embryogenesis while the zygotic genome remains transcriptionally silent. Developmental control is then passed from mother to zygote through a process known as the maternal-to-zygotic transition (MZT). The MZT comprises an intimately connected set of molecular events that mediate degradation of maternally deposited mRNAs and transcriptional activation of the zygotic genome. This essential developmental transition is conserved among metazoans but is perhaps best understood in the fruit fly, *Drosophila melanogaster*. In this article, we will review our understanding of the events that drive the MZT in *Drosophila* embryos and highlight parallel mechanisms driving this transition in other animals.

## 1. Introduction

To enable the development of a new, unique animal, the first stages of embryonic development require that the unified germ cells rapidly transit to a totipotent state. Maternally provided products play an essential role in driving this developmental transition and help to reprogramme the early embryonic genome. Subsequent activation of the zygotic genome and degradation of these maternally deposited products allows for the control of embryonic development to transition from the mother to the zygote. This maternal-to-zygotic transition (MZT) requires the coordination of multiple events, including remodelling of the mitotic division cycle, morphological changes, widespread transcriptional activation of the zygotic genome and degradation of a subset of maternal transcripts and proteins [1–5]. Recent data have shown that chromatin structure and the three-dimensional (3D) architecture of the chromosomes are highly dynamic during this developmental transition and may also play pivotal roles in reshaping the early embryonic programme [6–10].

While the dramatic developmental changes that occur during the MZT are largely conserved among animals, the timing of these events is variable. The MZT occurs over the course of several hours in rapidly developing species like worms, flies, frogs and fish [11]; meanwhile, in mammalian pre-implantation embryos, this transition takes days to complete [12]. In *Drosophila melanogaster*, the early nuclear divisions occur at an unprecedented rate of every 8–10 min. The MZT spans 13 rapid nuclear divisions that occur without cytokinesis, giving rise to a syncytium. At the 14th nuclear cycle, the division cycle slows dramatically, and the approximately 6000 nuclei become cellularized [13,14]. Similar to other eukaryotes, zygotic genome activation (ZGA) occurs gradually with a minor wave beginning at nuclear cycle 8 and a major wave during cycle 14 (figure 1a) [12,15]. *Drosophila* offer an attractive system to study the principal

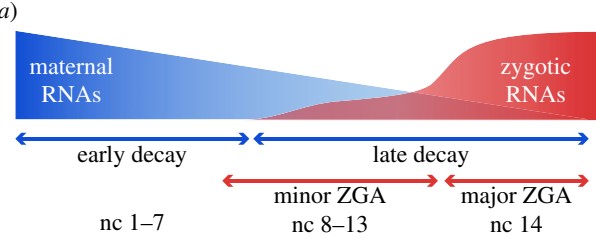

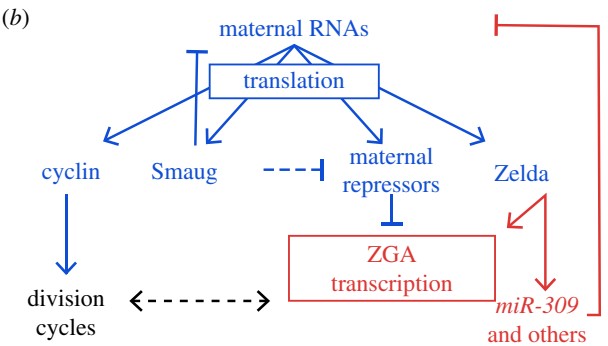

**Figure 1.** The interplay between maternal clearance and zygotic genome activation during the MZT in *Drosophila*. (*a*) Model of maternal and zygotic gene expression dynamics over the MZT. Maternal mRNA stores are eliminated through the action of two RNA degradation pathways: an 'early decay' pathway driven by maternally contributed factors following egg activation, independent of zygotic transcription; and a 'late decay' pathway directed by zygotically expressed factors. Zygotic genome activation (ZGA) occurs gradually over the MZT, with early onset expression of about one hundred zygotic genes (minor ZGA) appearing several nuclear cycles (nc) before the subsequent widespread activation of the zygotic genome (major ZGA). (*b*) Maternally loaded RNAs and proteins translated from these RNAs (blue) regulate molecular events governing the MZT, including mitotic division-cycle dynamics, maternal mRNA turnover and zygotic genome activation. Products of zygotic transcription (red), in turn, contribute to division-cycle remodelling and maternal RNA destabilization.

mechanisms that govern this highly conserved developmental transition because of their rapid development, well characterized genetic tools and the ability to harvest large numbers of precisely staged embryos.

In the following review, we discuss progress made towards understanding the regulatory principles directing the MZT in *Drosophila melanogaster*. We highlight how these findings have furthered our understanding of this critical developmental transition in other systems and the implications for cellular reprogramming.

# 2. Post-transcriptional dynamics of early development

Genome reprogramming was first successfully demonstrated by John Gurdon using somatic cell nuclear transfer (SCNT) of differentiated somatic cells into enucleated frog embryos [16]. SCNT was performed later in mammals with the iconic cloning of Dolly the sheep [17]. These seminal studies demonstrated that the oocyte contains all the factors needed to convert a differentiated cell to a naive developmental state and initiate new life. While reprogramming in culture is inefficient [18], during normal development a fertilized egg is transformed

into a totipotent zygote with high efficiency. Therefore, investigating how the maternal products within the egg control cell fate during the first stages of development will uncover molecular mechanisms that are required for rapid and efficient cellular reprogramming.

## 2.1. Maternal control initiates development

The first stages of embryogenesis occur in the absence of *de novo* transcription from the zygotic genome, and maternal products direct all cellular processes. Regulation in the early embryo is solely governed by post-transcriptional mechanisms, including those that regulate the translation, stability and subcellular localization of mRNAs. In *Drosophila*, approximately 55–65% of the genome is maternally contributed as RNA, and these maternal transcripts are essential for development [15,19–21].

Post-transcriptional regulation of maternally encoded RNAs controls protein expression during the first stages of embryonic development (figure 1*b*). This regulation is mediated, in part, through *cis*-acting elements within the deposited RNAs. These elements contain a combinatorial code targeted by RNA-binding proteins (RBPs) that regulate mRNA localization, translation and degradation [22]. Regulation of spatio-temporal expression of maternal mRNAs is necessary to establish developmental patterning in the embryo [22,23]. One of the best-characterized RBPs acting in the oocyte and early embryo is Staufen. First identified in *Drosophila* [24], Staufen is evolutionarily conserved and has a central role in mRNA transport, localization and translation [25–27]. Staufen is responsible for the localization of several essential maternal mRNAs, including *bicoid* and *oskar*, that define anterior and posterior axes of the embryo [24,25].

During these initial stages of development, translational efficiency is influenced by RBPs and poly(A) tail length. Elongation of the poly(A) tails of maternal mRNAs is developmentally regulated in many organisms, and this lengthening correlates with the increased translation of maternal mRNAs [28–31]. This coupling of poly(A) tail length and translational efficiency disappears after gastrulation and has yet to be observed at any other time in development [32]. In the early *Drosophila* embryo, translation of the essential maternal mRNA *smaug* (*smg*) is regulated by both poly(A) tail lengthening and alleviation of repression [19]. SMG itself is an RBP, which regulates stability and translation of targeted maternal transcripts [19,33]. Thus, translational control serves as one regulatory mechanism to help precisely time the events required to initiate development (figure 1*b*).

## 2.2. Clearance of the maternal instructions

Once maternal factors have initiated embryonic development, the maternal instructions—in the form of mRNAs and proteins—are actively cleared from the zygote to remove the previous cellular identity. Although there is no conclusive evidence demonstrating that maternal mRNA clearance is required during the MZT, it is thought to be a prerequisite for ZGA. In *Drosophila*, the combined activity of two general RNA decay pathways ensures the elimination of maternally deposited transcripts during the MZT (figure 1*a*) [34]. The first pathway is maternally encoded, in which maternally deposited factors initiate the degradation of maternal transcripts in the absence of fertilization and zygotic transcription. The second pathway is dependent on the expression of factors from the zygotic

genome that contribute to maternal RNA clearance late in the MZT [34,35]. The concerted action of both the 'maternal degradation' and 'zygotic degradation' pathways together allows for the clearance of necessary maternal mRNAs and for genetic control of development to be passed to zygotically synthesized products [34–36].

The early wave of maternally driven mRNA degradation accounts for clearance of over half of all maternal mRNAs degraded during the MZT [34,37]. This process is regulated in large part through RBPs that bind to *cis*-acting elements within the open reading frames or 3′-untranslated regions of maternal transcripts, including SMG, Brain tumour (BRAT) and Pumillio (PUM) [19,33,36,38,39]. While the mechanisms by which these factors cause RNA degradation are not fully understood, they function in part through recruitment of de-adenylases such as the CCR4/POP2/NOT complex [40–43]. Because SMG, PUM and BRAT largely have non-overlapping targets, collectively they bind to and direct the clearance of a significant subset of maternal mRNAs [5,33,36]. To ensure that maternal transcripts are not prematurely degraded during oogenesis, RNA decay in the early embryo must be triggered following egg activation. This is controlled, in part, by a kinase signalling cascade, which regulates expression of these RBPs. For example, the maternal mRNA encoding SMG is translationally repressed by PUM in the mature oocyte [19]. Following egg activation, the maternally supplied PAN GU (PNG) kinase is activated and relieves PUM-mediated repression, promoting *smg* mRNA translation. Together with maternally contributed piRNAs, SMG facilitates deadenylation and degradation of hundreds of maternal mRNAs (figure 1*b*) [19,44–46].

With the onset of zygotic transcription at 1–2 h after fertilization, additional mRNA degradation mechanisms are initiated [34]. One major contributor to this zygotically expressed maternal mRNA decay pathway is the *miR-309* cluster that contains eight microRNA (miR) genes (figure 1*b*) [47]. This cluster of zygotically expressed genes is required for the degradation of approximately 400 maternal mRNAs [47]. Similarly in zebrafish, *miR-430* is one of the first transcribed zygotic RNAs and functions to regulate the stability of hundreds of maternal mRNAs [48]. Likewise, *Xenopus miR-427* accumulates very early during the MZT and mediates destabilization of maternal mRNAs encoding cyclins [49]. Thus, microRNA-mediated mRNA degradation is a conserved mechanism regulating maternal mRNA degradation during the MZT in many species. Despite the conservation and importance of these microRNAs, multiple genomic regions are required for maternal mRNA clearance, suggesting many factors involved in the zygotic degradation pathway have yet to be identified [20]. Together, these clearance pathways allow for the maternal programme to be erased and coupled to the activation of transcription from the zygotic genome.

## 3. Regulation of the division cycle

In many organisms, the initial stages of development are characterized by a series of rapid cellular divisions without significant growth leading to the generation of multiple totipotent cells. During *Drosophila* embryogenesis, the first 2 h of development comprise 13 rapid nuclear division cycles within a shared cytoplasm (figure 2*a*). These mitotic cycles lack gap phases and are therefore a series of repeating cycles of DNA synthesis and mitosis with each cycle occurring approximately every 10 min [13,14]. During these rapid cycles, completion of S-phase is rate limiting to cycle progression [50]. The onset of nuclear cycle 14 initiates several major developmental changes essential for completing the MZT: cellularization of the nuclei, prolonged DNA replication leading to a lengthened S-phase, introduction of G2 phase and widespread ZGA (figure 2*a*).

Abundant maternal factors direct the rapid early embryonic cleavage cycles, in which the entire genome is replicated simultaneously, and gap phases are bypassed to hasten the division cycle. As development progresses, DNA replication takes longer and longer to complete largely due to the onset of late-replicating sequences at satellite and heterochromatic regions. Delays in DNA replication lengthen S-phase, and in cycle 14 a pause between the completion of DNA synthesis and mitosis occurs with the introduction of G2 gap phase. This division cycle pausing in G2 requires the inhibition of maternally deposited drivers of the division cycle, cyclin and cyclin-dependent kinase 1 (Cdk1), to block mitotic entry [51–55].

Cyclin/Cdk1 activity not only controls mitotic cycling (figure 1*b*) but is also associated with the delay in DNA replication timing that occurs during the 14th division cycle. During the early rapid cycles, synchronous replication of late-replicating satellite sequences with the rest of the genome is dependent on cyclin/Cdk1 activity [51]. Initial slowing of the division cycle and DNA replication in the blastoderm stages coincides with maternal cyclin destruction and Cdk1 inactivity. Mechanistically, recent data show that the developmental downregulation of cyclin/Cdk1 permits the binding of maternal protein Rap1-interacting factor 1 (Rif1) at satellite regions [56]. Rif1 is a candidate repressor of replication thought to mediate the introduction of late-replicating satellite sequences [56]. With the onset of late replication, a DNA replication checkpoint response delays mitosis to ensure time to complete DNA synthesis [14,50,57]. While early replicating nuclei undergo mitosis without this surveillance mechanism, DNA replication checkpoint genes and S-phase elongation are essential for proper execution of the final nuclear cycles of the MZT [50,57,58].

Onset of zygotic transcription also influences the DNA replication checkpoint. Sites of stalled DNA replication overlap with actively transcribed loci, and reducing nascent zygotic transcription is sufficient to avoid conflicts between the DNA replication machinery and RNA polymerase [59]. By contrast, simply blocking RNA polymerase II elongation was insufficient to bypass a functional DNA replication checkpoint. Thus, acting upstream of productive transcription, the establishment of accessible regions of the genome occupied by the active or poised polymerase is thought to cause replication stalling and checkpoint activation [59]. These findings imply that ZGA together with replication stress triggers division-cycle slowing (figure 1*b*) [59,60]. As a result, remodelling of the division cycle during the MZT is controlled by multiple overlapping mechanisms, including S-phase elongation, inhibition of cyclin/Cdk1, introduction of late DNA replication and checkpoints, and transcriptional activation of the genome.

Because transcription is limited during mitosis, the abbreviated S phases during early development only permit a brief period of transcriptional competence per cycle. Based on RNA polymerase elongation rates and the short time allowed for transcription during these early cycles, small transcript size and highly efficient splicing are likely prerequisites for early expressed genes in many organisms [13,14,61–64].

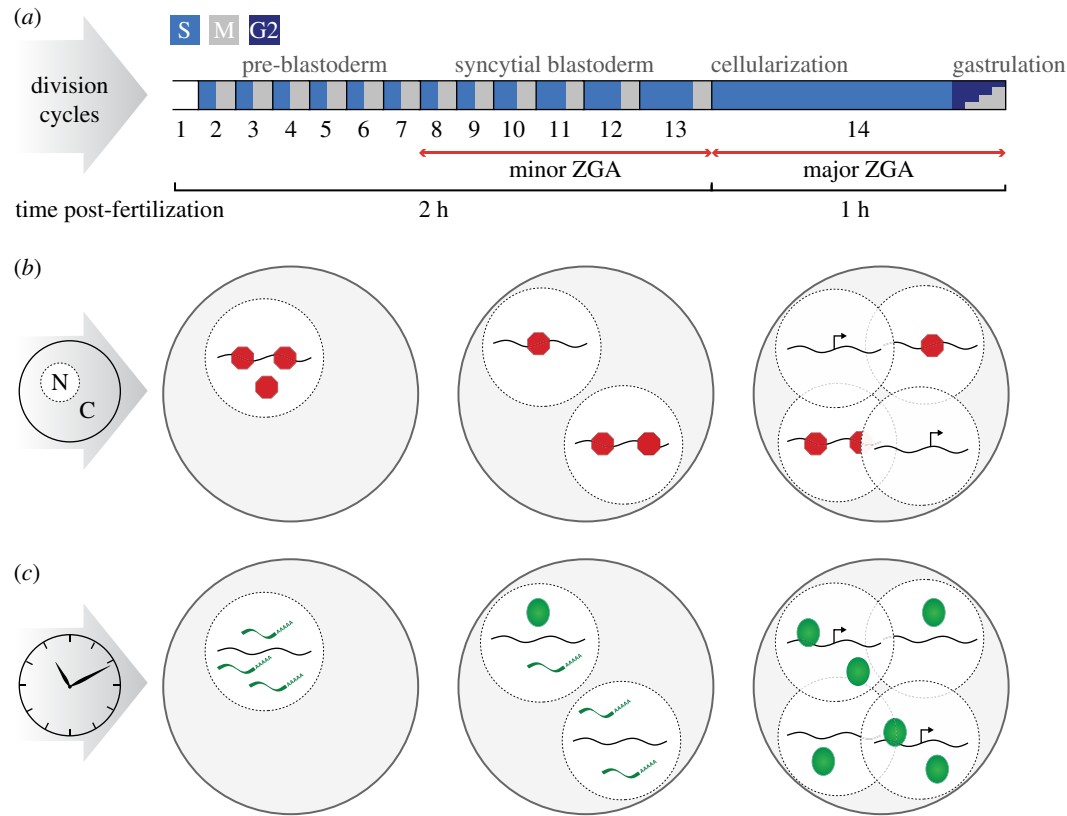

**Figure 2.** Multiple mechanisms trigger the onset of division-cycle remodelling and zygotic transcription. (*a*) The early embryo exists as a syncytium of nuclei undergoing rapid division cycles of repeated DNA replication (S) and mitosis (M). Progressive elongation of S-phase permits time to achieve transcriptional competence from the zygotic genome. The major wave of genome activation occurs at the onset of cycle 14, accompanied by cellularization of nuclei and the introduction of a gap phase (G2). (*b*) A proposed maternally supplied repressor (red) inhibits transcription in the early embryo. As the number of nuclei increases exponentially with each division, the nuclear : cytoplasmic ratio increases, titrating away the repressor and allowing zygotic transcription to initiate. (*c*) Maternal clock model in which translation of a maternal activator (green) requires a set amount of developmental time following fertilization to accumulate sufficient levels of protein to trigger ZGA.

Disproportionate to the genome average, the early expressed zygotic genes of flies [20], mosquitoes [63] and zebrafish [64] are short and intronless, while long zygotically expressed genes are expressed later during the *Drosophila* MZT relative to their shorter counterparts [61]. In this way, mitotic-cycle dynamics and transcript length influence the activation of zygotic transcription (figure 1*b*).

# 4. Zygotic genome activation

In nearly all metazoans, ZGA does not initiate immediately following fertilization. Instead, ZGA is coordinated with the degradation of maternal products and slowing of the division cycle. During *Drosophila* embryogenesis, ZGA occurs within the first few hours following fertilization. Multiple mechanisms contribute to the temporal regulation of ZGA and allow activation to be precisely coupled with the other processes that take place during this essential developmental transition. Ultimately, their interconnectedness ensures a smooth handoff from a maternally driven programme to nascent zygotic control during the MZT.

## 4.1. Coordinating ZGA with other cellular processes

Many cellular processes must be coordinated to allow progression through the MZT, including division-cycle slowing and activation of the zygotic genome (figure 2*a*). Two non-

mutually exclusive biological regulatory mechanisms have been proposed to facilitate this coordination: (i) changes in the nuclear to cytoplasmic ratio (N : C) within the syncytial embryo, and (ii) the time elapsed since fertilization, referred to as a 'maternal clock' (figure 2*b*,*c*) [59].

The embryos of many species, including *Drosophila* and humans, do not change in volume during the initial stages of development. Thus, while each round of DNA replication and division amplifies the number of nuclei exponentially, the volume of the cytoplasm remains unchanged leading to a progressive change in the ratio of nuclear DNA to cytoplasmic components. Manipulation of the N : C ratio through changes in zygotic ploidy can uncover the relative contribution of the two mechanisms: N : C ratio verses 'maternal clock'. For example, haploid embryos require an additional round of nuclear replication to achieve the same N : C ratio as a diploid [65,66]. These embryos have half the amount of DNA relative to diploids and undergo one additional division cycle prior to cellularization [53,65,66]. Conversely, triploid embryos carrying one and a half the DNA content of a diploid undergo one fewer division [67]. Therefore, the number of division cycles is responsive to the N : C ratio and can be adjusted accordingly to ensure the correct nuclear density upon cellularization.

The N : C ratio has also been shown to influence the onset of zygotic gene expression in several species [66,68–72], leading to the theory that the transition from transcriptional quiescence to transcriptional activity is a result of titrating away a

maternally loaded repressor with the increasing zygotic DNA content (figure 2b). In haploid embryos, the expression of dozens of zygotic genes is delayed and some of these genes act to inhibit division-cycle progression [52,66]. Thus, the N : C ratio serves as a mechanism to synchronize changes to the division cycles with changes to the zygotic transcriptome.

By contrast, expression of many other zygotic genes is independent of the N : C ratio and depends on alternative mechanisms, including the time-dependent 'maternal clock', to regulate the timing of ZGA (figure 2c). Cyclohexamide-mediated inhibition of translation prior to nuclear cycle 10 prevents ZGA [65], suggesting that translation of certain factors are essential for the onset of zygotic gene expression. One possible time-dependent factor is the maternal transcription factor Zelda (ZLD). Maternally deposited zld mRNA is translationally upregulated at approximately cycle 10 [73] and is essential for zygotic gene expression [19]. Nevertheless, while the molecular mechanisms asserting the 'maternal clock' remain elusive, it is evident that the combined action of both the N : C ratio and time-dependent factors help to co-ordinate developmental events necessary to execute the MZT with precision.

## 4.2. Zelda is a master regulator of ZGA

The Drosophila transcription factor ZLD was the first-identified master regulator of ZGA in any organism [74]. zld is required as a maternally deposited mRNA that is translationally upregulated following fertilization, and this translational control may be essential for timing ZGA [73–75]. Translational upregulation of transcription factors may be a general mechanism to control the timing of ZGA. Increases in expression of Pou5f3, Sox19b and Nanog in zebrafish and TATA-binding protein in Xenopus contribute to the onset of ZGA in these species [76–78]. Thus, transcriptional activation of the zygotic genome may occur when a threshold of activating factors is reached.

ZLD binds to a class of related DNA-sequence motifs, termed TAGteam elements, that are enriched in the cis-regulatory elements of genes activated during ZGA [79]. At nuclear cycle 10, ZLD occupies 64% of the canonical TAGteam motifs (CAGGTAG), and this set of ZLD-bound loci remains largely unchanged as ZGA progresses [80]. Genes with the highest ZLD occupancy in their promoter and enhancer regions correspond to the initial genes transcribed during the MZT (nuclear cycles 8–10) [80]. It is likely that high levels of ZLD binding to promoters and proximal enhancers is sufficient to directly activate a subset of zygotic genes as the addition of ZLD-binding motifs can drive precocious gene expression during the MZT [79]. In addition, increased ZLD activity results in the upregulation of target genes [81]. Nonetheless, thousands of early ZLD-bound regions are not active until the major wave of ZGA (nuclear cycle 14) [80]. At these loci, ZLD acts to poise genes for expression. ZLD functions, in part, to establish or maintain regions of open chromatin, which potentiates the binding of other transcription factors [6,7].

ZLD-target genes are some of the earliest transcribed genes during the MZT. Among the genes that require ZLD for early expression are those involved in sex determination, pattern formation and cellularization [74,79]. To ensure the embryo survives past the blastoderm stage, these cellular processes must be initiated prior to widespread gene expression [82,83]. Additionally, ZLD directly activates components of the zygotic

RNA degradation pathways that destabilize maternal RNAs, including the miR-309 cluster (figure 1b) [47,74,84]. In this way, the activation of zygotic transcription is coordinated with the degradation of maternally deposited RNAs, allowing a handoff in developmental control from mother to embryo. The role of ZLD in activating the zygotic genome is likely not unique to Drosophila as functional data combined with phylogenetic analysis supports a role for ZLD during the MZT in other insects and crustaceans [63,79,81,85–90].

Maternally deposited zld encodes a 1596 amino acid protein that contains six C2H2 (Cys–Cys–His–His motif) zinc fingers [74,75,88]. In tissue culture, ZLD is a robust transcriptional activator and requires a C-terminal cluster of four zinc fingers for DNA binding and a low-complexity sequence for activating transcription [74,88]. In addition to the DNA-binding domain, ZLD contains additional highly conserved protein domains, including an acidic patch and two C2H2 zinc fingers within the N-terminus [81,89]. ZLD activity is partially controlled through the second conserved N-terminal zinc finger (ZnF2), as mutation of this domain results in a hyperactive mutant [81]. Deletion of maternally provided zld, overexpression, or hyperactivation via mutations in the ZnF2-inhibitory domain all result in embryonic lethality [74,75,81]. Therefore, ZLD levels and activity must be precisely controlled for the embryo to properly navigate the MZT.

Although ZLD was first identified as an essential activator of the zygotic genome over a decade ago [74,75], the mechanism by which ZLD functions is still unknown. Nonetheless, a variety of experiments have begun to elucidate features by which ZLD activates the zygotic genome. These experiments have shown that ZLD exhibits many characteristics of a pioneer transcription factor in vivo [91,92]. DNA wrapped around histone proteins forms nucleosomes that package the DNA into chromatin. In vivo, transcription factors must access their binding motifs within the context of chromatin and license the genome for activation. Pioneer factors are a unique class of transcription factors that bind to nucleosomal DNA, establish regions of accessible chromatin and allow for the recruitment of additional factors [91]. In Drosophila, enhancer regions of early expressed zygotic genes are thought to have an intrinsically high nucleosome barrier [7], and ZLD overcomes this barrier through local depletion of nucleosomes near ZLD motifs [6,7]. The fact that ZLD binding is driven largely by sequence suggests that, similar to pioneer factors, DNA incorporation into nucleosomes is not a barrier to ZLD binding or that the chromatin of the early embryo is widely accessible [80]. Furthermore, ZLD is one of the first factors to engage target sites in chromatin prior to zygotic gene activity [80] and this binding establishes or maintains chromatin accessibility to permit the binding of other factors (figure 3a,b) [6–8,93–96]. ZLD modulates not only transcription-factor binding but also the timing and strength of enhancer activity in response to maternal morphogen gradients [93]. Collectively, these data demonstrate that ZLD serves as a master regulator of ZGA—regulating early zygotic gene expression, transcription-factor binding and chromatin accessibility in the early embryo.

Recent breakthroughs in fixed and live-cell imaging have provided insights into how ZLD and other pioneering factors may function. Confocal and lattice-light sheet imaging have shown that ZLD forms dynamic subnuclear hubs in the pre-blastoderm embryo [97,98]. Bicoid similarly forms transient hubs of local high density in the posterior of the early Drosophila embryo, and these hubs are dependent on ZLD [99]. Thus, to

rsob.royalsocietypublishing.org   Open Biol. 8: 180183

rsob.royalsocietypublishing.org    Open Biol. **8**: 180183

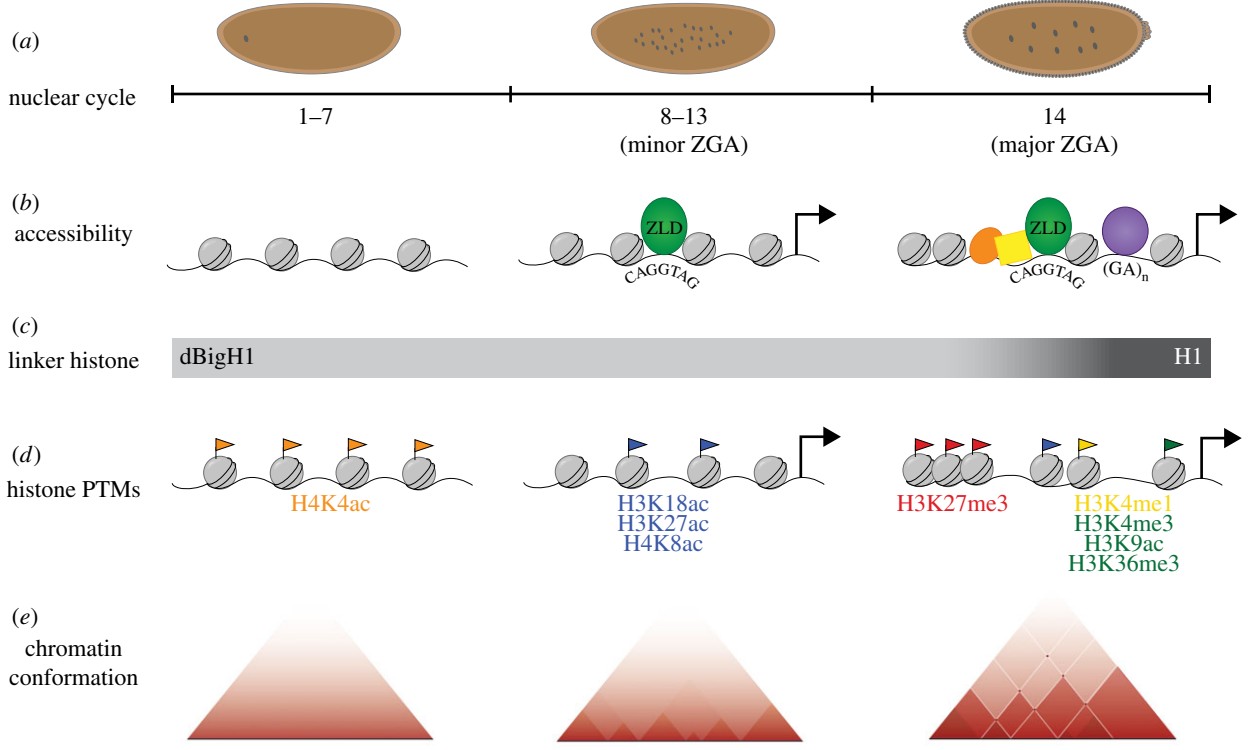

**Figure 3.** Dynamic changes in the chromatin landscape correlate with zygotic genome activation. (*a*) Schematic of staged embryos at nuclear cycles spanning early, mid and late MZT. (*b*) Maternal ZLD is required to maintain regions of open chromatin near early expressed zygotic genes. Later, during the major wave of ZGA, both binding sites for ZLD (green) and GA-di-nucleotide binding proteins (purple) are enriched at accessible regions of the genome. (*c*) Embryonic linker histone variant, dBigH1, is abundant in the early embryo when the genome is inactive. During cycles 13–14, coinciding with the major wave of ZGA, dBigH1 is replaced with somatic H1. (*d*) As the MZT progresses, there is an overall increase in histone modifications incorporated into the zygotic genome. H4K5ac is highest during early nuclear cycles associated with rapid DNA replication. At cycles 8–12, histone acetylation marks (blue) are enriched near loci activated during the minor wave of ZGA. During cycles 13–14, there is an increase in both histone methylation marks associated with active (green) and repressive (red) transcription. (*e*) Progressive demarcation of topologically associated domain (TAD) boundaries, as determined by Hi-C data, occurs over the MZT concomitant with ZGA.

overcome the combination of the low concentration of Bicoid in the posterior of the embryo and low-affinity binding to chromatin, ZLD hubs may concentrate Bicoid, and possibly other transcription factors, at specific genomic loci and in so doing facilitate factor binding and subsequent gene expression. The formation of such hubs is not unique to the early embryo and has been reported for several other factors in *Drosophila* and in human embryonic stem cells [100–104]. These data suggest that master-regulatory transcription factors may function in part to scaffold assemblages of proteins with diverse activities to facilitate rapid and coordinated gene expression profiles.

## 4.3. Transcriptional regulation by maternal factors collectively reprogrammes the zygotic transcriptome

Regions of the genome characterized by early onset expression correlate with regions that require ZLD for chromatin accessibility [6,8]. However, this association is not found for zygotic genes expressed later during the major wave of ZGA. Thus, while ZLD is required to activate hundreds of genes during the ZGA, additional factors play a role in mediating widespread genome activation. Regions of open chromatin that remain accessible even in the absence of ZLD are enriched for GA di-nucleotide repeats (figure 3*a,b*) [6,7]. These same regions are also enriched in regions of the genome that gain chromatin accessibility and RNA polymerase II pausing during the major wave of ZGA [8]. The enrichment of these

motifs in *cis*-regulatory regions suggests proteins that bind to this sequence are instrumental in ensuring the activation of the zygotic genome. Two proteins expressed from maternally deposited mRNAs, GAGA factor (GAF) and chromatin-linked adaptor for MSL proteins (CLAMP), bind to these GA di-nucleotide repeats and both have been implicated in ZGA [105–107]. While these proteins can compete for binding, they have specificity for different sequence motifs [108] and may have non-overlapping roles during ZGA. GAF has many functions throughout development and is required for robust transcription and nuclear divisions during the MZT [109,110]. Similarly, maternally provided CLAMP is essential for early development [107]. CLAMP is recruited to the histone locus body and is required for both promoting chromatin accessibility and activating zygotic expression of all replication-dependent histone genes during ZGA [107]. Although the specific roles of CLAMP and GAF during ZGA remain to be elucidated, they are likely to have essential functions. In addition to these two factors, additional maternal and zygotic transcription factors, including STAT92E [111], are likely to function during this conserved transition. Thus, the collective action of multiple factors allows for ZGA to be precisely executed.

In other organisms, multiple transcription factors and epigenetic regulators have been identified that influence ZGA. Recent studies have demonstrated that it is likely not the action of a single master regulator that controls ZGA, but instead a network of factors each regulating specific transcriptional or chromatin changes. In zebrafish, three transcription factors—Pou5f3, Sox2 and Nanog—together drive the minor

rsob.royalsocietypublishing.org Open Biol. 8: 180183

wave of ZGA [77,78]. In mammals, the transcription factors Stella, YAP1, Nfya and Oct4 are all maternally deposited, but each has been found to activate only a subset of zygotic genes [112–117]. The early expressed zygotic factor DUX/DUX4 is important for activating ZGA-related genes in mammalian cleavage stage embryos [118,119], but how it itself is initially expressed is unclear. Mechanistically, factors in flies (ZLD, GAF and CLAMP) [6–8], zebrafish (Nanog, Pou5f3, SoxB1 and Cohesin) [77,78,120,121], mice (Nfya and Dux) [115,118,119,122], and humans (DUX4 and OCT4) [117–119,122] have all been shown to drive the early developmental programme and mediate chromatin accessibility. Therefore, the collective action of multiple transcriptional regulators is likely a conserved mechanism by which the genome is remodelled to regulate zygotic gene expression.

# 5. Chromatin dynamics and epigenetic modifications reshape the embryonic genome

The cellular reprogramming that occurs during the initial stages of development as the fertilized egg transitions to the totipotent cells of the early embryo is driven by a monumental shift in the transcriptome of the organism. This process is accomplished with the simultaneous reprogramming of the early embryonic genome through changes to the histone content, post-translational modifications (PTMs) of histone proteins and global changes in chromatin architecture (figure 3).

## 5.1. Histone exchange during the MZT

Maternal and paternal genomes are differentially packaged in each respective gamete, and nuclear reprogramming of the parental genomes is required for activation of the resulting zygotic genome following fertilization. Nuclear remodelling begins with rapid changes to the paternal genome, in which the sperm nucleus is transformed into the male pronucleus. Paternal DNA is initially packaged primarily with sperm-specific protamines in place of histones [123,124], enabling the formation of highly compacted chromatin that is not permissive for replication [125]. At fertilization, protamines are expelled and replaced by maternal histones resulting in the rapid remodelling of the sperm chromatin. This process is highly conserved and is largely controlled by maternal factors present in the egg [124–126].

Over the MZT, there are dramatic changes to the chromatin structure of both the maternal and paternal genome beyond the exchange of protamines for histones in the male pronucleus. The importance of this remodelling is reflected in the fact that one of the earliest events during SCNT is the exchange of somatic histones with oocyte-specific variants [127–129]. This mirrors early embryonic development in which most metazoans replace germ-cell-specific histone variants for their somatic counterparts. Among histone proteins, the linker histone H1 class is the most divergent [130]. Several species have specific germline H1 variants that play specific roles during gametogenesis and embryogenesis [131,132]. H1 regulates transcription and embryogenesis by influencing chromatin compaction, higher-order chromatin structure and heterochromatic silencing [133,134]. H1 variants display different affinities for chromatin, a characteristic that may help diversify chromatin structure in specific developmental contexts.

Drosophila have a single identified maternal-specific H1 variant that is present during the MZT, dBigH1 [135]. Similar to events during SCNT, incubation of somatic nuclei in Drosophila preblastodermal embryo extract induced dBigH1 incorporation into chromatin [136]. dBigH1 is essential for gametogenesis and early embryonic development. Prior to cellularization, dBigH1 is uniformly associated with chromatin and prevents premature ZGA. dBigH1 is replaced by somatic H1 upon cellularization but remains in the quiescent primordial germ cells into late embryogenesis (figure 3c) [135]. These data suggest that the replacement of embryonic variant dBigH1 with somatic H1 renders the genome transcriptionally competent at the time of ZGA. Similar to Drosophila dBigH1, maternal linker histone H1 variants are retained in the oocytes of frogs, zebrafish and mammals and persist during the early stages of embryogenesis [137–141]. While no unifying functional properties between these oocyte-specific H1 variants have been determined, the fact that maternal linker histones replace somatic H1 in the germline and early embryo in a wide range of metazoans suggests a role in facilitating totipotency.

## 5.2. Changes in post-translational histone modifications over the MZT

Studies in many organisms demonstrate that reprogramming of the early embryonic genome is characterized by the loss of most chromatin modifications, a transient period of unmodified or naive chromatin and the subsequent re-establishment of defined chromatin states. During the MZT, there is a dramatic increase in the abundance of histone modifications incorporated into the zygotic genome, and the timing of this increase coincides with the onset of ZGA [142–146]. While in Drosophila zygotic gene expression initiates at cycle 8, at this time in development these early expressed regions are devoid of H3K4me3, a post-translational modification associated with transcriptionally active promoters [145,147]. Additional marks commonly associated with active genes, H3K9ac and H3K36me3, also do not appear until cycle 14 [145], suggesting that these marks are not required for transcriptional competence. By contrast, enhancers and promoters of the earliest expressed zygotic genes are already enriched for a subset of histone acetylation marks (H3K18, H3K27 and H4K8) as early as cycle 8 and these modifications increase genome-wide as the embryo progresses through ZGA (figure 3d) [145]. These modifications may be important for ZLD-mediated genome activation, as the canonical ZLD-binding motif is highly correlated with the position of these initial histone acetylation marks, and histone acetylation is decreased in embryos lacking maternal zld [145]. Furthermore, H3K18ac and H4K8ac are enriched around loci that require ZLD for chromatin accessibility [6]. While the exact mechanism by which histone acetylation is established in the early Drosophila embryo remains unknown, acetylation may be a functionally conserved mechanism for activating the zygotic genome. Recently, the histone acetyltransferase p300/CBP and the histone acetyl binding protein BRD4 were shown to be instrumental in genome activation during the MZT in zebrafish, and increased histone acetylation resulted in premature activation of zygotic transcription [148].

Additional histone modifications associated with active chromatin, such as H3K9ac, H3K4me1 and H3K4me3, become enriched at specific genomic loci during the final cycles of the MZT (figure 3*d*) [145]. Polycomb-mediated H3K27me3 also increases at this time although low levels have been reported immediately following fertilization [145,147,149]. The increase in both activating and repressing histone modifications through the MZT suggests that these marks are required only as the zygotic genome becomes transcriptionally active. While global levels of histones increase only slightly over the MZT, as measured by H3 occupancy, histone methylation and marks of facultative heterochromatin become significantly enriched near the end of the MZT [145]. The accumulation of specific chromatin signatures likely signifies the demarcation of an increasingly structured or defined chromatin state as the embryo transitions through the MZT.

## 5.3. Chromatin dynamics and architecture

The eukaryotic genome is partitioned into topological and functionally distinct active and repressed domains [150]. Consisting of several layers of higher-order structures, the three-dimensional conformation of the genome within the nucleus is complexly organized, highly dynamic and important for regulating gene expression [151,152]. Topologically associated domains (TADs) confine and insulate certain regulatory regions of the genome, structurally organizing interphase chromatin [153]. In mammalian genomes, TADs encompass between 200 kb and 2 Mb [154,155], while domains in flies are comparatively smaller, being on average 60 kb [152,156]. Chromatin conformation capture methods reveal chromosome organization within the nucleus and have helped to determine the physical proximity of genes, chromatin loops and enhancer–promoter interactions [157,158]. Recent data suggest that the zygotic genome exists in a largely unstructured state prior to ZGA, and that, in addition to the increase in histone modifications, the chromatin architecture similarly becomes more organized with the onset of zygotic transcription [9,10,117,146,159].

In *Drosophila* embryogenesis, hierarchical compartmentalization of the genome occurs in multiple waves (figure 3*e*). Active chromatin loops and early TAD boundaries are first evident during the minor wave of ZGA. Nonetheless, these chromosomal interactions are weak and are rapidly followed by another structural transition correlating with the major wave of ZGA that further demarcates active and repressive chromatin domains [9,10]. Although correlated with the onset of transcription from the zygotic genome, TAD formation in *Drosophila* embryos occurs independent of transcription, as higher-order structural TADs remain intact even when transcription is pharmacologically inhibited [9]. Moreover, topological boundaries are similar between anterior and posterior sections of the embryo despite having differences in spatio-temporal transcriptional profiles during the MZT [159]. Conversely, density within TADs and inter-TAD contacts are affected by active transcription [9]. This signifies that while TAD formation as a whole is transcription independent, insulation between TADs and overall organization within individual domains is dependent on transcription.

Despite recent advances in our understanding of TAD formation during early embryogenesis, it remains unclear what initially nucleates and guides chromatin folding during early development. Topological boundaries established during the MZT are enriched in histone acetylation marks, chromatin accessibility and ZLD occupancy [9]. During this time, ZLD is required to induce long-range chromatin looping at early activated zygotic genes and helps define a subset of TAD boundaries [9,10]. Given the role of ZLD in determining chromatin accessibility, it is likely that ZLD, additional transcription factors and insulator proteins cooperatively direct changes in chromatin conformation. Indeed, data indicate that factors such as the transcription factor GAF, Polycomb group proteins (PcG) and other chromatin-binding proteins also function to remodel chromatin and establish TAD boundaries during the MZT [6,9,10].

TADs appear to be common within eukaryotic genomes and are often conserved among cell types and even across closely related species [160–162], suggesting a conserved means to control genome function. As observed in fly embryogenesis, there is an absence of chromatin structure within mouse embryos and TADs gradually form following ZGA [163,164]. Genome organization in developing zebrafish embryos differs slightly such that strong compartmentalization and TAD boundaries are present immediately following fertilization, and the genome does not lose structural features until ZGA [165]. ZGA is not essential for the formation of TADs in flies, zebrafish or mice [9,164,165]. However, an interesting finding from mouse studies revealed that the establishment of TADs during the MZT is DNA replication dependent [164]. As comparable studies have not yet been performed in flies, it remains unclear whether this is a conserved requirement. Together, these studies identify a transient period during early embryogenesis in which unstructured chromatin accompanies reprogramming of the metazoan genome. Despite the emerging literature on reprogramming of higher-order chromatin structure in the early embryo, it remains unclear how chromatin loops and TADs contribute to the regulation of gene expression throughout development. Future studies investigating the causality of 3D genome reprogramming and gene expression will further our understanding of the regulatory principles driving totipotency.

## 6. Concluding remarks

Decades of insights from *Drosophila melanogaster* embryogenesis have paved the way towards a better understanding of the MZT and the dramatic transcriptional and cellular changes that govern this developmental transition. As the embryo transitions from fertilized egg to totipotent zygote, the embryonic genome and transcriptome need to be rapidly reprogrammed. This rapid and efficient *in vivo* embryonic transformation has a number of similarities with the much less efficient reprogramming of specified cell types to induced pluripotent stem cells in culture [18]. This parallel is made clear by the fact that the core pluripotency factors Pou5f3, Sox2 and Nanog drive ZGA during the initial stages of zebrafish embryonic development [77,78], and Oct4 in mammalian pre-implantation embryos [116,117]. Although these proteins are not conserved in *Drosophila* at the sequence level, they function in a similar manner to ZLD during early embryogenesis. They all share the capacity to facilitate chromatin accessibility and reprogramme cells to a naive developmental state, and misexpression of these factors can be detrimental to the cell or organism. Thus, in many species, a network of maternal transcription factors acts through conserved mechanisms to reprogramme the early

rsob.royalsocietypublishing.org Open Biol. **8**: 180183

embryonic genome. How additional factors and mechanisms cooperate with essential transcription factors to collectively reprogramme the genome during the MZT remains an important area for future research.

Given that division-cycle regulation, DNA replication, transcriptional activation, maternal RNA clearance and chromatin remodelling are all tightly coupled during the MZT, an exciting future challenge will be elucidating the mechanisms that coordinate these essential developmental processes. The advancements of single-cell sequencing technologies and high-resolution microscopy have allowed the critical events driving this developmental transition to be understood in ever greater detail. Through our elucidation of these conserved developmental mechanisms, we will provide important insights into the unifying principles that drive genome activation.

Data accessibility. This article has no additional data.

Competing interests. We declare we have no competing interests.

Funding. Work in the Harrison lab was supported by grants from the National Institutes of General Medical Sciences (R01GM111694), the American Cancer Society (RSG DDC-130854) and a Vallee Scholar Award.

Acknowledgements. We thank members of the Harrison lab for helpful feedback, as well as the reviewers of this manuscript.

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
