## [Reviewer comments · Open Biology]

Review History

RSOB-18-0183.R0 (Original submission)

Review form: Reviewer 1

Recommendation

Accept with minor revision (please list in comments)

Are each of the following suitable for general readers?

- a) **Title**
Yes
- b) **Summary**
Yes
- c) **Introduction**
Yes

Is the length of the paper justified?

Yes

Should the paper be seen by a specialist statistical reviewer?

No

Is it clear how to make all supporting data available?

Not Applicable

Is the supplementary material necessary; and if so is it adequate and clear?

Not Applicable

Do you have any ethical concerns with this paper?

No

Comments to the Author

Review of Open Biology Manuscript RSOB-18-0183 by Leila Rieder

Hamm & Harrison: Insights from *Drosophila melanogaster*: regulatory principles governing the maternal-to-zygotic transition

Summary:

Hamm and Harrison have written a very comprehensive yet succinct review focusing on MZT in the *Drosophila* embryo. They provide comparisons, when available, to other species, and outline the layers of change--from transcription factors to PTMs to high order genome organization- that occurs when maternally deposited factors hand off development to the zygotic genome. The figures are excellently rendered and clear.

Major criticisms:

Consider eliminating passive voice.

One of the goals of the review is to illustrate how the cells of the early embryo achieve totipotency, however I find this a bit lacking. For example, there is no discussion of the protamine-to-histone conversion of the male pronucleus. I suggest this as an alternative to the discussion on SCNT, which I found out of place, or perhaps as a paragraph in the beginning of section 5.

Consider adding a paragraph about sex determination, since the sex determination cascade involves zygotic factors and begins in the embryo prior to MZT.

Minor Criticisms:

Please add nuclear cycle numbers to Figure 1A

Please mention the conservation of Staufén (page 4, line 71).

Be specific when writing "transcriptional activation of the zygotic genome," for example, page 5, line 115. Is this the earlier wave of activation (NC8) or the major wave (NC14)?

Figure 3B shows Zelda binding to a TAGteam motif that is not nucleosomal--I suggest adding a nucleosomal motif to the figure.

Typos and Grammar:

Add the word "to" on page 5, line 125.

If convention is not to italicize "*Drosophila*," then do not italicize "*Xenopus*" (page 7, line 230)

Remove "the" from page 7 line 241

Some acronyms do not need to be noted, as they are not used again.

"3D" on page 8, line 272 should be "3B" referring to the figure, or else the statement needs clarification.

“Figure 3C” on page 10 line 374 should be “Figure 3D.”

Add a “.” to page 12 line 468

“Less” on page 6, line 204 should be “fewer.”

Review form: Reviewer 2

Recommendation

Accept with minor revision (please list in comments)

Are each of the following suitable for general readers?

- a) **Title**
Yes
- b) **Summary**
Yes
- c) **Introduction**
Yes

Is the length of the paper justified?

Yes

Should the paper be seen by a specialist statistical reviewer?

No

Is it clear how to make all supporting data available?

Not Applicable

Is the supplementary material necessary; and if so is it adequate and clear?

Not Applicable

Do you have any ethical concerns with this paper?

No

Comments to the Author

The manuscript by Hamm and Harrison describe the molecular mechanisms that underlie the MZT, with a focus on what we have learned from studies in *Drosophila*. The review is timely, readable and comprehensive.

Below are a number of minor points the authors should consider prior to publication.

Minor points:

Inverting the title might make sense (i.e. Regulatory principles governing the maternal-to-zygotic transition: Insights from *Drosophila melanogaster*)

While the degradation of maternal mRNAs is thought to be required during the MZT, there is no evidence that conclusively demonstrates this. The text should be modified to reflect this.

Drosophila should be italicized throughout.

Clarify what is meant by 'division cycle' in line 24.

For clarity lines 61+62 should be modified to read: 'including those that regulate the translation, stability and subcellular localization of mRNAs'.

Lines 65+66 should be modified to read 'through cis-acting elements within the deposited RNAs' to reflect the fact that elements can be found in UTRs as well as in orfs.

The sentence in lines 75+76 as worded could be read so as to suggest that polyA tail length is regulated independent of RBPs, which is in fact likely not the case.

Line 82. SMG regulates both the stability and translation of target mRNAs (i.e. Chen et al., 2014 Genome Biology).

Lines 86+92 should be modified to indicate that there are two 'general' pathways of decay to recognize that there are likely multiple maternal and zygotic pathways.

Lines 99-100. While PUM and BRAT binding sites are largely found in 3'UTRs, SMG sites are largely located in orfs (see Chen et al., 2014 Genome Biology). In addition, while SMG functions maternally, BRAT appears to function in both maternal and zygotic pathways (Laver et al, 2015 Genome Biology), while PUM appears to function in zygotic pathways (see Thomsen et al., 2010, Genome Biology and Laver et al., 2015 Genome Biology).

The role of piRNAs in mRNA decay is very controversial, and to avoid the authors having to delve into this controversy, I suggest that they remove references to this from their manuscript.

Lines 229 and 230: Should this sentence read 'Increases in expression of Pou5f3, Sox19b, Nanog in zebrafish and TATA-binding protein in *Xenopus* contributes to the onset of ZGA in these species'?

Lines 233-238. Its unclear what the relationship is between genes bound by ZLD and those genes that required ZLD for their activation. Are the later all direct targets or are some fraction indirect? Also, where are the binding sites in the targets typically found?

Delete 'clear' from line 208.

Decision letter (RSOB-18-0183.R0)

29-Oct-2018

Dear Dr Harrison

We are pleased to inform you that your manuscript RSOB-18-0183 entitled "Insights from *Drosophila melanogaster*: regulatory principles governing the maternal-to-zygotic transition" has been accepted by the Editor for publication in Open Biology. The reviewer(s) have recommended publication, but also suggest some minor revisions to your manuscript. Therefore, we invite you to respond to the reviewer(s)' comments and revise your manuscript.

Please submit the revised version of your manuscript within 14 days. If you do not think you will be able to meet this date please let us know immediately and we can extend this deadline for you.

- 1) A text file of the manuscript (doc, txt, rtf or tex), including the references, tables (including captions) and figure captions. Please remove any tracked changes from the text before submission. PDF files are not an accepted format for the "Main Document".
- 2) A separate electronic file of each figure (tiff, EPS or print-quality PDF preferred). The format should be produced directly from original creation package, or original software format. Please note that PowerPoint files are not accepted.
- 3) Electronic supplementary material: this should be contained in a separate file from the main text and meet our ESM criteria (see <http://royalsocietypublishing.org/instructions-authors#question5>). All supplementary materials accompanying an accepted article will be treated as in their final form. They will be published alongside the paper on the journal website and posted on the online figshare repository. Files on figshare will be made available approximately one week before the accompanying article so that the supplementary material can be attributed a unique DOI.

Online supplementary material will also carry the title and description provided during submission, so please ensure these are accurate and informative. Note that the Royal Society will not edit or typeset supplementary material and it will be hosted as provided. Please ensure that the supplementary material includes the paper details (authors, title, journal name, article DOI). Your article DOI will be 10.1098/rsob.2016[last 4 digits of e.g. 10.1098/rsob.20160049].

- 4) A media summary: a short non-technical summary (up to 100 words) of the key findings/importance of your manuscript. Please try to write in simple English, avoid jargon, explain the importance of the topic, outline the main implications and describe why this topic is newsworthy.

Images

Data-Sharing

It is a condition of publication that data supporting your paper are made available. Data should be made available either in the electronic supplementary material or through an appropriate repository. Details of how to access data should be included in your paper. Please see <http://royalsocietypublishing.org/site/authors/policy.xhtml#question6> for more details.

Data accessibility section

Sincerely,

The Open Biology Team
<mailto:openbiology@royalsociety.org>

Reviewer(s)' Comments to Author:

Referee: 1

Comments to the Author(s)

Review of Open Biology Manuscript RSOB-18-0183 by Leila Rieder

Hamm & Harrison: Insights from *Drosophila melanogaster*: regulatory principles governing the maternal-to-zygotic transition

Summary:

Hamm and Harrison have written a very comprehensive yet succinct review focusing on MZT in the *Drosophila* embryo. They provide comparisons, when available, to other species, and outline the layers of change--from transcription factors to PTMs to high order genome organization--that occurs when maternally deposited factors hand off development to the zygotic genome. The figures are excellently rendered and clear.

Major criticisms:

Consider eliminating passive voice.

One of the goals of the review is to illustrate how the cells of the early embryo achieve totipotency, however I find this a bit lacking. For example, there is no discussion of the protamine-to-histone conversion of the male pronucleus. I suggest this as an alternative to the discussion on SCNT, which I found out of place, or perhaps as a paragraph in the beginning of section 5.

Consider adding a paragraph about sex determination, since the sex determination cascade involves zygotic factors and begins in the embryo prior to MZT.

Minor Criticisms:

Please add nuclear cycle numbers to Figure 1A

Please mention the conservation of Staufen (page 4, line 71).

Be specific when writing “transcriptional activation of the zygotic genome,” for example, page 5, line 115. Is this the earlier wave of activation (NC8) or the major wave (NC14)? Figure 3B shows Zelda binding to a TAGteam motif that is not nucleosomal—I suggest adding a nucleosomal motif to the figure.

Typos and Grammar:

Add the word “to” on page 5, line 125.

If convention is not to italicize “*Drosophila*,” then do not italicize “*Xenopus*” (page 7, line 230)

Remove “the” from page 7 line 241

Some acronyms do not need to be noted, as they are not used again.

“3D” on page 8, line 272 should be “3B” referring to the figure, or else the statement needs clarification.

“Figure 3C” on page 10 line 374 should be “Figure 3D.”

Add a “.” to page 12 line 468

“Less” on page 6, line 204 should be “fewer.”

Referee: 2

Comments to the Author(s)

The manuscript by Hamm and Harrison describe the molecular mechanisms that underlie the MZT, with a focus on what we have learned from studies in *Drosophila*. The review is timely, readable and comprehensive.

Below are a number of minor points the authors should consider prior to publication.

Minor points:

Inverting the title might make sense (i.e. Regulatory principles governing the maternal-to-zygotic transition: Insights from *Drosophila melanogaster*)

While the degradation of maternal mRNAs is thought to be required during the MZT, there is no evidence that conclusively demonstrates this. The text should be modified to reflect this.

Drosophila should be italicized throughout.

Clarify what is meant by ‘division cycle’ in line 24.

For clarity lines 61+62 should be modified to read: ‘including those that regulate the translation, stability and subcellular localization of mRNAs’.

Lines 65+66 should be modified to read ‘through cis-acting elements within the deposited RNAs’ to reflect the fact that elements can be found in UTRs as well as in orfs.

The sentence in lines 75+76 as worded could be read so as to suggest that polyA tail length is regulated independent of RBPs, which is in fact likely not the case.

Line 82. SMG regulates both the stability and translation of target mRNAs (i.e. Chen et al., 2014 *Genome Biology*).

Lines 86+92 should be modified to indicate that there are two ‘general’ pathways of decay to recognize that there are likely multiple maternal and zygotic pathways.

Lines 99-100. While PUM and BRAT binding sites are largely found in 3'UTRs, SMG sites are largely located in orfs (see Chen et al., 2014 Genome Biology). In addition, while SMG functions maternally, BRAT appears to function in both maternal and zygotic pathways (Laver et al, 2015 Genome Biology), while PUM appears to function in zygotic pathways (see Thomsen et al., 2010, Genome Biology and Laver et al., 2015 Genome Biology).

The role of piRNAs in mRNA decay is very controversial, and to avoid the authors having to delve into this controversy, I suggest that they remove references to this from their manuscript.

Lines 229 and 230: Should this sentence read 'Increases in expression of Pou5f3, Sox19b, Nanog in zebrafish and TATA-binding protein in *Xenopus* contributes to the onset of ZGA in these species'?

Lines 233-238. Its unclear what the relationship is between genes bound by ZLD and those genes that required ZLD for their activation. Are the later all direct targets or are some fraction indirect? Also, where are the binding sites in the targets typically found?

Delete 'clear' from line 208.

Author's Response to Decision Letter for (RSOB-18-0183.R0)

See Appendix A.

Decision letter (RSOB-18-0183.R1)

09-Nov-2018

Dear Dr Harrison

We are pleased to inform you that your manuscript entitled "Regulatory principles governing the maternal-to-zygotic transition: insights from *Drosophila melanogaster*" has been accepted by the Editor for publication in Open Biology.

Sincerely,

The Open Biology Team
mailto: openbiology@royalsociety.org

Appendix A

Thank you for the opportunity to write this review and to respond to the helpful comments from the reviewers. We appreciate that they found the review "comprehensive yet succinct" and "timely, readable, and comprehensive." Below you will find our point-by-point response to each comment.

Referee: 1

Comments to the Author(s)

Review of Open Biology Manuscript RSOB-18-0183 by Leila Rieder

Hamm & Harrison: Insights from *Drosophila melanogaster*: regulatory principles governing the maternal-to-zygotic transition

Summary:

Hamm and Harrison have written a very comprehensive yet succinct review focusing on MZT in the *Drosophila* embryo. They provide comparisons, when available, to other species, and outline the layers of change--from transcription factors to PTMs to high order genome organization- that occurs when maternally deposited factors hand off development to the zygotic genome. The figures are excellently rendered and clear.

Major criticisms:

Consider eliminating passive voice.

Thank you for this suggestion. We have reviewed the manuscript with this in mind.

One of the goals of the review is to illustrate how the cells of the early embryo achieve totipotency, however I find this a bit lacking. For example, there is no discussion of the protamine-to-histone conversion of the male pronucleus. I suggest this as an alternative to the discussion on SCNT, which I found out of place, or perhaps as a paragraph in the beginning of section 5.

We have added a brief discussion of the histone to protamine exchange at the beginning of this section.

Consider adding a paragraph about sex determination, since the sex determination cascade involves zygotic factors and begins in the embryo prior to MZT.

We appreciate the importance of sex determination and its establishment in the early embryo. While entire reviews have been devoted to this process, we have made a brief mention of the fact that genes involved in this pathway and in early patterning comprise the first group of expressed genes in the early embryo.

Minor Criticisms:

Please add nuclear cycle numbers to Figure 1A

These have been added.

Please mention the conservation of Staufen (page 4, line 71).

We have mentioned that Staufen is conserved, but given the scope of this review have chosen to focus on it's role in *Drosophila*.

Be specific when writing "transcriptional activation of the zygotic genome," for example, page 5, line 115. Is this the earlier wave of activation (NC8) or the major wave (NC14)?

For the specific expression of *mir-309* the exact timing of expression has not been reported. We have changed the wording for clarity and looked for additional examples throughout the text where existing data would allow us to be more precise in our description of gene expression timing.

Figure 3B shows Zelda binding to a TAGteam motif that is not nucleosomal--I suggest adding a nucleosomal motif to the figure.

While we and Chris Rushlow have unpublished data demonstrating that Zelda can bind to nucleosomal DNA, there are no published data to warrant inclusion in Figure 3B.

Typos and Grammar:

Add the word "to" on page 5, line 125. **Incorporated edit as suggested.**

If convention is not to italicize "*Drosophila*," then do not italicize "*Xenopus*" (page 7, line 230) **Italicized *Drosophila* throughout.**

Remove "the" from page 7 line 241 **Incorporated edit as suggested.**

Some acronyms do not need to be noted, as they are not used again.

“3D” on page 8, line 272 should be “3B” referring to the figure, or else the statement needs clarification. **Incorporated edit as suggested.**

“Figure 3C” on page 10 line 374 should be “Figure 3D.” **Incorporated edit as suggested.**

Add a “.” to page 12 line 468 **Incorporated edit as suggested.**

“Less” on page 6, line 204 should be “fewer.” **Incorporated edit as suggested.**

Referee: 2

Comments to the Author(s)

The manuscript by Hamm and Harrison describe the molecular mechanisms that underlie the MZT, with a focus on what we have learned from studies in *Drosophila*. The review is timely, readable and comprehensive.

Below are a number of minor points the authors should consider prior to publication.

Minor points:

Inverting the title might make sense (i.e. Regulatory principles governing the maternal-to-zygotic transition: Insights from *Drosophila melanogaster*) **The title has been inverted, as suggested.**

While the degradation of maternal mRNAs is thought to be required during the MZT, there is no evidence that conclusively demonstrates this. The text should be modified to reflect this.

We have modified the text to reflect the lack of conclusive evidence for the requirement of maternal mRNA degradation..

Drosophila should be italicized throughout. **This is now italicized throughout.**

Clarify what is meant by ‘division cycle’ in line 24. **Corrected to read, “mitotic division cycle”.**

For clarity lines 61+62 should be modified to read: ‘including those that regulate the translation, stability and subcellular localization of mRNAs’. **Thanks for the suggestion. The text has been changed as suggested.**

Lines 65+66 should be modified to read ‘through cis-acting elements within the deposited RNAs’ to reflect the fact that elements can be found in UTRs as well as in orfs. **We agree and have changed the text to be more inclusive as suggested.**

The sentence in lines 75+76 as worded could be read so as to suggest that polyA tail length is regulated independent of RBPs, which is in fact likely not the case. **We appreciate pointing out the possible lack of clarity and have attempted to revise it to be clearer.**

Line 82. SMG regulates both the stability and translation of target mRNAs (i.e. Chen et al., 2014 Genome Biology). **We have incorporated and cited work from Chen et al., 2014.**

Lines 86+92 should be modified to indicate that there are two ‘general’ pathways of decay to recognize that there are likely multiple maternal and zygotic pathways. **This has been modified.**

Lines 99-100. While PUM and BRAT binding sites are largely found in 3’UTRs, SMG sites are largely located in orfs (see Chen et al., 2014 Genome Biology). In addition, while SMG functions maternally, BRAT appears to function in both maternal and zygotic pathways (Laver et al, 2015 Genome Biology), while PUM appears to function in zygotic pathways

(see Thomsen et al, 2010, Genome Biology and Laver et al., 2015 Genome Biology). **The text has been edited to include regulation through both 3'UTRs and ORFs.**

The role of piRNAs in mRNA decay is very controversial, and to avoid the authors having to delve into this controversy, I suggest that they remove references to this from their manuscript. **We appreciate this suggestion and for this reason have chosen to keep our mention of this role to a minimum, but provide references. This will allow readers to come to their own conclusions.**

Lines 229 and 230: Should this sentence read 'Increases in expression of Pou5f3, Sox19b, Nanog in zebrafish and TATA-binding protein in Xenopus contributes to the onset of ZGA in these species'? **These changes have been made.**

Lines 233-238. Its unclear what the relationship is between genes bound by ZLD and those genes that required ZLD for their activation. Are the later all direct targets or are some fraction indirect? Also, where are the binding sites in the targets typically found?

We have added a discussion to clarify the relationship between ZLD-bound loci and gene expression.

Delete 'clear' from line 208. **Modified line 308.**